# The Temperature Field Evolution and Water Migration Law of Coal under Low-Temperature Freezing Conditions

**DOI:** 10.3390/ijerph182413188

**Published:** 2021-12-14

**Authors:** Bo Li, Li Li, Laisheng Huang, Xiaoquan Lv

**Affiliations:** 1School of Safety Science and Engineering, Henan Polytechnic University, Jiaozuo 454003, China; anquanlili2021@163.com (L.L.); anquanhls@163.com (L.H.); shizhen202105@163.com (X.L.); 2Collaborative Innovation Center of Coal Work Safety and Clean High Efficiency Utilization, Henan Polytechnic University, Jiacozuo 454003, China; 3State Key Laboratory Cultivation Base for Gas Geology and Gas Control, Henan Polytechnic University, Jiaozuo 454003, China

**Keywords:** low-temperature freezing, temperature propagation, freezing front, water migration, coal

## Abstract

This study examines the evolution law of the coal temperature field under low-temperature freezing conditions. The temperature inside coal samples with different water contents was measured in real-time at several measurement points in different locations inside the sample under the condition of low-temperature medium (liquid nitrogen) freezing. The temperature change curve was then used to analyse the laws of temperature propagation and the movement of the freezing front of the coal, which revealed the mechanism of internal water migration in the coal under low-temperature freezing conditions. The results indicate that the greater the water content of the coal sample, the greater the temperature propagation rate. The reasons for this are the phase change of ice and water inside the coal during the freezing process; the increase in the contact area of the ice and coal matrix caused by the volume expansion; and the joint action of the two. The process of the movement of the freezing front is due to the greater adsorption force of the ice lens than that of the coal matrix. Thus, the water molecules adsorbed in the unfrozen area of the coal matrix migrate towards the freezing front and form a new ice lens. Considering the temperature gradient and water content of the coal samples, Darcy’s permeation equation and water migration equation for the inside of the coal under freezing conditions were derived, and the segregation potential and matrix potential were analysed. The obtained theoretical and experimental results were found to be consistent. The higher the water content of the coal samples, the smaller the matrix potential for the hindrance of water migration. Furthermore, the larger the temperature gradient, the larger the segregation potential, and the faster the water migration rate.

## 1. Introduction

The thermal effect of the freezing of rock via ambient temperatures has long been a popular issue in rock thermodynamic theory and engineering research. Geotechnical construction in cold regions is currently on the rise, e.g., the construction of bridges, tunnelling, mining, the cryogenic geological storage of liquefied natural gas and liquefied petroleum gas, and construction by the freezing method, all of which subject the surrounding rock to periodic or long-term freezing at low temperatures. Therefore, from an engineering perspective, the study of the temperature field evolution of coal bodies under low-temperature freezing conditions is of great significance for engineering construction in cold regions. The water migration that occurs in the coal rock during or after freezing is of interest because it changes, to varying degrees, the phase change of the water in the coal rock and the resulting stresses, as well as the intensity and rate of displacement. The migration of unfrozen water from the coal rock to the frozen surface that occurs during the freezing process produces freezing and deformation, which cause hazards to foundation works, pit support, mine excavation, and tunnelling. In addition, the study of the migration of unfrozen water provides theoretical support for the design and construction of freezing projects and the application of the freezing method to actual projects. It also provides a valuable reference for the treatment and prevention of frost damage to projects, and is of great significance for the safety and economic efficiency of projects. Moreover, the study of coal rocks during the freeze–thaw process due to water redistribution can provide greater theoretical guidance due to the microstructural occurrence and the resulting changes in the strength and deformation characteristics. The study of water migration inside coal under cryogenic freezing conditions is very important to prevent gas disasters and cryogenic freezing damage. Liquid nitrogen cryogenic freezing technology is a very environmentally friendly technology with little risk to the environment and human health. The study of the internal water migration of coal under liquid nitrogen cryogenic freezing technology can also reduce environmental hazards and benefit public health. Anthracite coal was chosen for the experiment, which has a hard texture, and the experimental results can be applied to the same type of coal. Experimental results can also be applied for materials with similar mechanical properties to the experimentally selected coal.

In recent years, both domestic and international scholars have carried out experimental and numerical simulations of the evolution of rock temperature fields under low-temperature freezing conditions. For instance, Park et al. [1] experimentally studied the thermophysical parameters of rocks in relation to temperature; when the temperature was varied between 40 and 160 °C, the specific heat and thermal expansion coefficient of the rock decreased with the decrease in temperature, whereas the thermal conductivity did not change substantially. To investigate the heat transfer pattern of rocks during freeze–thawing, Kenji et al. [2] tested and calibrated five different sensors on the market. Shen et al. [3] and McDermott et al. [4] monitored temperature changes at different locations within a sandstone sample during freezing by embedding a temperature sensor in the sandstone. Guo et al. [5] placed temperature sensors inside rock samples and investigated the temperature equilibrium pattern of rocks during freeze–thawing via a combination of experimental and numerical analyses. Zhang et al. [6] applied the finite element method to the study of thermal-poroelasticity to develop a numerical model that considers the phase change of the pore water during freezing, and to predict the temperature transfer during freezing. Neaupane et al. [7] conducted non-linear elastoplastic simulations of the freeze–thawing of rock and compared the simulation results with experimental results to demonstrate the accuracy of the simulations in the prediction of temperature transfer during freezing. Vitelv et al. [8] used heat transfer theory and numerical simulations to explore the characteristics of the effect of frozen pipes on the temperature of the surrounding rock. Lunardini [9] proposed a “three-zone” mathematical model to more realistically describe the rock freeze–thaw process, suggested that the soil system will exhibit certain phase change zones during the temperature evolution process, and provided a solution to solve for the three-zone temperature propagation. Walder [10] and others have investigated the ways in which rock pores freeze, crack, and expand, but none have elucidated the mechanism of water migration within the pore space. Sudisman et al. [11] experimentally investigated the relationship between hydraulic conductivity and the development of frozen temperature fields in natural soils, demonstrating that water migration has an effect on the development of temperature fields. Chen et al. [12] found that during freezing periods, water in porous media such as rock and concrete tends to form ice or migrate, leading to the redistribution of pore water. Li et al. [13,14,15,16] studied the freeze–thaw effect of liquid nitrogen and the pore size, porosity, and permeability of coal rocks; these characteristics were all found to increase with the increase of water saturation. One study analysed the effects of the changes in different conditions on water migration by developing a coupled hydrothermal model for numerical simulation [17]. Nagare et al. [18] conducted two-way freezing tests to investigate the effect of temperature on the soil water potential and water content redistribution; the process of moisture migration was found to be the migration of unfrozen water to the frozen peak surface. Tan et al. [19] and Taron et al. [20] divided the freezing process into frozen and unfrozen zones based on the location of the frozen peak surface, and used the variable substitution method to analyse the temperature field. Based on the results of 141 differential scanning calorimetry (DSC) experiments conducted on six single-mineral soils, Kozlowski [21] presented a semi-empirical model describing the variation of the unfrozen water content with the temperature in a frozen soil-water system. While these studies have good guidance value for the heat transfer law of coal rocks under low-temperature conditions, they were largely focused on theoretical derivation and numerical simulation, and were characterised by fewer experimental studies and a lack of experimental data. Most of these scholars studied the temperature propagation and water migration mechanisms of rocks and soils under low-temperature freezing conditions, but there have been fewer corresponding experimental studies on coal, resulting in inadequate experimental data and theoretical analyses related to the temperature propagation and water migration mechanisms of coal under low-temperature freezing conditions. Moreover, most of the experimental protocols adopted the temperature gradient as a single variable and considered only a single influencing factor, thereby ignoring the influences of other factors.

To address these shortcomings, this study focuses on the evolution of the temperature field of the coal under low-temperature freezing conditions based on laboratory tests, and the water content and the distance of the measurement point inside the coal from the cold source are considered as variables. Ultimately, the process of moisture migration during freezing is revealed. This research was conducted to provide solutions to various engineering problems, such as the exposure of the surrounding rock of engineering structures to periodic or long-term low-temperature freezing conditions, and to provide a theoretical basis for the construction of cold-zone projects. Moreover, this study provides a valuable reference for the treatment and prevention of frost damage to projects, and also has important significance for the safety and economic benefits of projects.

## 2. Coal Sample Preparation and Experimental Methods

### 2.1. Coal Sample Preparation

The anthracite coal from coal seam II-1 of the Zhaogu No. 2 mine in Jiaozuo City, Henan Province, China, was considered as the object of this research. Fresh bulk coal samples taken from underground were selected, and the raw coal was processed into a number of cylindrical coal samples with dimensions of Φ50 × 140 mm using a core drilling machine and a core cutting machine with a tolerance range of ±1 mm. To reduce the influence of the differences between the coal samples on the temperature measurement results, the coal samples were all sourced from the same large piece of raw coal. Five boreholes with a diameter of 5 mm were drilled evenly above the prepared coal samples to different depths (40, 60, 80, 100, and 120 mm). By controlling the distance between drill holes at around 10 mm, it ensures that the holes are completely independent of each other under cooling conditions, as shown in Figure 1.

The procedure for the preparation of coal samples with different water saturation levels used in the test was as follows.
(1)The processed coal sample was placed in a drying oven and dried at a constant temperature of 60 °C. The sample was weighed until the weight no longer decreased, and the weight was recorded as the dry mass of the sample.(2)The coal sample was treated with full water using a vacuum water device, during which the sample was weighed every 12 h until the weight no longer increased and the sample was considered saturated. The mass of the saturated water sample was recorded.(3)The saturated coal sample was placed in the drying oven, during which it was continuously removed and weighed (the weighing time was adjusted according to the actual needs) until the target dry mass was reached. The sample was then removed and immediately placed in a sealed bag in which it was naturally cooled to room temperature. The coal sample was placed in a constant-temperature and -humidity cabinet for moisture balance, the temperature was set to room temperature (25 °C), and the coal sample was weighed every 4 h during the process of moisture balance. The process generally took 3–5 days. The coal sample was weighed repeatedly until its weight was nearly unchanged, which is regarded as the completion of moisture balance.(4)Steps (1)–(3) were repeated to prepare coal samples with different water saturation levels.

It was presupposed that the water content saturation was, respectively, 0, 50%, and 100%, and the target dry mass was calculated as follows:(1)$$m_{i}=S_{i}(m_{s}-m_{d})+m_{d}$$
where $m_{i}$ is the target dry mass,$\;S_{i}$ is the preset water saturation, $m_{s}$ is the mass of the saturated water sample, and $m_{d}$ is the mass of the dried coal sample.

### 2.2. Experimental System

The equipment used in this experiment was a self-developed real-time temperature measurement device for self-pressurised liquid nitrogen cold-soaking. The experimental system included a self-pressurised liquid nitrogen tank, a liquid nitrogen insulation container, a coal sample holding device, a temperature sensor, and a real-time temperature acquisition device, as shown in Figure 2. The liquid nitrogen holding vessel in which the cooling was performed during the experiments was completely sealed, so that no external air could enter the interior. The measurement range of the temperature measurement instrument in this test device was −200 to 200 °C, and the measurement accuracy was ±0.1 °C. The temperature sensor was a three-wire PT100A-grade platinum resistor (Figure 3), the probe diameter was 4 mm, and the probe length was 100 mm; moreover, the temperature measurement range was −200 to 100 °C, and the error when reaching −100 to −200 °C was not more than 0.5 °C. The temperature measurement resistor of the sensor was located at the top of the probe, thereby allowing for the precise measurement of the temperature at a single point. The liquid nitrogen used in the experiment had a purity of 99.99% and a boiling point of −195.8 °C at 0.1 MPa. Thus, the experimental device was able to achieve the temperature measurement effect.

### 2.3. Experimental Methods and Process

To study the effect of the movement of the freezing front of coal rock on the temperature field under low-temperature freezing conditions, the water content and the distance from the cold source were considered as variables to perform temperature measurement tests in real time at different measurement points inside the coal. The experimental protocol was as follows.

(1) To study the temperature propagation of the coal under low-temperature freezing conditions, tests were carried out at different distances (20, 40, 60, 80, and 100 mm) from the cold source. To prevent the liquid nitrogen and the cryogenic nitrogen produced by the liquid nitrogen from penetrating the coal sample wall along the holder during the experiment, a silicone ring was placed between the bottom of the coal sample and the holder. The liquid nitrogen was placed in contact with the bottom of the coal sample to ensure that the temperature propagation started from the bottom of the coal sample. The coal sample was also held tightly in place.

(2) To investigate the effect of the water content on the temperature field of the coal and the evolution of the freezing front, the temperature variation inside the coal sample at different distances (20, 40, 60, 80, and 100 mm) from the cold source was monitored at different water content saturations (0%, 50%, and 100%).

The specific experimental steps were as follows. (I) Controlling the depth of each hole to within plus or minus 1 mm and the diameter of the hole to within plus or minus 0.1 mm, to ensure that the location and area of the sensor contact point in each hole was the same. After testing and commissioning of the experimental equipment, all temperature sensors were placed in the borehole of the coal sample to be measured and sealed in a liquid nitrogen insulated container. The sealed liquid nitrogen insulation container was left at room temperature for 2–3 h to ensure that the initial temperatures of the sensors and the coal samples were the same, and then the experiment was conducted. (II) The self-pressurizing liquid nitrogen tank was opened so that it injected liquid nitrogen into the holding tank; the temperature collection interval was 1 min, during which the liquid nitrogen was intermittently injected into the holding tank, and the liquid nitrogen surface was always kept just submerged at the bottom of the coal sample. (III) When the temperature variation at each point was less than 0.01 °C/min, the temperature field of the coal sample was considered to have reached equilibrium, and the data collection was halted. (IV) The coal sample was removed, and the next set of tests was carried out.

The present experiment is a further study based on previous work; the relevant replicate experiments have been performed in the previous work.

## 3. Results and Discussion

### 3.1. Effect of the Water Content on the Temperature Propagation Pattern of the Coal

Water will phase into ice when in contact with a cold source, and the thermal conductivity of ice and water differ significantly; thus, moisture is an important factor in the propagation of temperature within the coal rock mass. To study the effect of the water content on the internal temperature propagation pattern of the coal, real-time temperature experiments were carried out on the coal via the self-developed device presented in Figure 2. The time curves of the temperature variation of different locations inside the coal samples with different water contents under the same experimental conditions were obtained, as exhibited in Figure 4.

Figure 4a–c, respectively, exhibit the temperature change at different measurement points inside the coal samples with water contents of 0, 4.23%, and 8.76%. It can be seen that with the increase in freezing time, the temperature of each measurement point inside the coal sample decreased continuously and finally reached a stable value. Moreover, the rate of temperature change was distributed in three stages, namely, the early, middle, and late freezing stages. At the end of the freezing period, the rate of temperature at each measurement point within the sample gradually became zero, and the final temperature was close to the stable value. However, the rate of temperature change at each measurement point varied; the closer the location to the cold source, the faster the rate of temperature change, the shorter the time required for the temperature field to reach relative steady state.

A comparison of Figure 4a–c reveals that although the temperature field changes of coal samples with different water contents exhibited the same patterns, there were obvious differences; thus, the temperature change curves of the coal samples with different water contents at the location of 60 mm from the cold source were compared and analysed, as shown in Figure 5. As the water content of the coal increased, the faster the rate of temperature change, the shorter the time required for the temperature field to reach a relative steady state; however, when frozen for long enough, the temperature at which this measurement point finally experienced equilibrium did not change with the change in water content. This reveals that as the water content of the coal increased, the rate of temperature change gradually increased, and the time required for the temperature field to reach a relative steady state decreased.

Considering that when the temperature changes through the freezing range of water, some deviations in the curve occur based on the phase change of moisture. Based on Figure 5, the time variation pattern of the three coal samples reaching a certain temperature at a fixed distance was explored by the temperature variation curve from 4 °C to −10 °C, as shown in Figure 6. The specific temperature variation with time is shown in Table 1.

It can be found that it takes 4 min to reduce the temperature of the measurement point from 4 °C to −10 °C for the coal sample with an 8.76% water content, 6 min for the coal sample with a 4.23% water content, and 9 min for the coal sample with a 0% water content. Again, it shows that the rate of temperature change of the coal gradually increases as the water content increases. The detailed area of the temperature change curve from 4 °C to −10 °C was further analysed by dividing the time required to reduce the temperature of the coal samples with different water contents from 4 °C to −10 °C into three parts of equal time: pre, mid, and post. As shown in Table 2, it can be found that coal samples with an 8.76% water content and coal samples with a 4.23% water content decreased more and more in the same time with the freezing process; especially, the temperature of coal samples with an 8.76% water content decreased the most in the later period when the temperature was below 0 °C, and the coal sample with a 0% water content decreased less and less in the same time with the freezing process. Comparative analysis of the variation pattern of the temperature reduction of coal samples with different water contents indicates that the phase change of water in the freezing range promotes the temperature propagation.

From the perspective of ice–water phase change, the decrease in the temperature of the coal in contact with a cold source causes the internal pore water to freeze, and the different thermal conductivities of ice and water affect the temperature propagation pattern of the coal. Some studies have shown that the thermal conductivity of ice is about four times that of liquid water at the same temperature, and the lower the temperature, the greater the thermal conductivity of ice [22]. The temperature of liquid nitrogen is extremely low, approximately −196 °C, so when the water phase of the pores inside the coal becomes ice, its thermal conductivity continues to increase, thereby accelerating the rate of temperature propagation through the coal. On the other hand, when water freezes, its volume increases with the phase change [23], and the pore water phase inside the coal becomes ice, resulting in volume expansion and the filling of the pores; the number of voids will then be relatively reduced, and the ice inside the pores will directly contact the coal matrix for heat transfer, thereby further accelerating the temperature propagation rate inside the coal. From the perspective of chemical potential, the chemical potential of solid-phase ice is lower than that of liquid-phase water [24]. Water always flows from a place of high chemical potential to a place of low chemical potential, thereby driving capillary water to migrate towards the fractionated ice within the pore space and accelerating the rate of temperature propagation. So, the phase change of water in the freezing range promotes the temperature propagation.

### 3.2. Movement Pattern of the Freezing Front of Coal Bodies under Different Water Content Conditions

As determined from the study of aqueous systems of bentonite with different salt concentrations [25], the phase change in water below −10 °C disappears when the salt concentration is zero. In the modelling of a clay water system [21], it was found that unfrozen water remains stable at −12 °C, but may be influenced by other factors, and the unfrozen water content will continue to decrease below −12 °C. In contrast, regarding the freezing process of pore water within rocks, the critical radius of the frozen pores no longer changes significantly at temperatures below −20 °C, and unfrozen water in pores smaller than the critical radius remains stable [26]. The results of uniaxial and triaxial compression tests of different rocks at 20 to −20 °C and dry saturation revealed an increase in the values of the uniaxial and triaxial compressive strengths, Young’s modulus, cohesion, and the friction angle with the decrease in the test temperature; however, no significant change was found at −20 °C and below [27,28]. A previous study of coal bodies under freezing conditions revealed that the changes in the uniaxial compressive strength, the elastic modulus, and the unfrozen water content of coal samples are no longer significant when the freezing temperature is below −20 °C [29]; it was found that when the temperature of coal decreases to −20 °C, the pore water of the coal can be considered to be completely frozen.

The temperature after 4 °C was divided into equal spacing, and the spacing was set to 14 °C. Under the same distance (60 mm), the time required to lower the same temperature for different water content coal samples was investigated, as shown in Table 3. It can be found that when the temperature range is from 4 °C to −24 °C, the time required to lower the same temperature becomes shorter for the coal sample with an 8.76% water content and 4.23% water content, and the time required to lower the same temperature remains the same for the coal sample with a 0% water content. When the temperature range is from −24 °C to −52 °C, the time required to lower the same temperature becomes longer for the coal sample with an 8.76% water content and the coal sample with a 4.23% water content, and the time required to lower the same temperature remains the same for the coal sample with a 0% water content. It means that when the temperature is higher than −20 °C, the pore water is continuously frozen into ice, which accelerates the rate of temperature propagation, so the time required to lower the same temperature of coal samples becomes shorter. When the temperature is lower than −20 °C, the pore water is regarded as completely frozen, so the time required to lower the same temperature of the coal sample becomes longer. When the temperature is lower than −52 °C, the time required to lower the same temperature for all three different water content coal samples gradually becomes longer. This indicates that the temperature gradient decreases gradually as the freezing process proceeds, and the time required to lower the same temperature for different water content coal samples becomes longer and longer, and when the freezing time is long enough, they will eventually lower to an identical temperature. Before that, the coal samples with different water contents will reach a relatively stable temperature and last for a long time, reaching a relatively stable state.

Thus, the isothermal surface of −20 °C was defined as the freezing front in the present study. The time taken to reach −20 °C was obtained from the temperature change curve at each measurement point within the coal, the movement pattern of the freezing front of the coal with the increase of the freezing time, as shown in Figure 7.

According to Figure 7, it can be seen that the distance travelled by the freezing front shows an increasing relationship with the growth of freezing time, first rapidly increasing and then gradually increasing slowly. When the coal body touches the freezing cold source, heat transfer between the coal body and the cold source takes place and freezing starts. At the early stage of freezing, the temperature gradient is large, so the freezing rate grows rapidly within a short period of freezing, and the freezing front moves rapidly to the other end of the cold source; at the later stage of freezing, the temperature gradient decreases, so the freezing rate starts to decrease, and the freezing front moves at a lower speed; finally, when the temperature inside the coal body reaches a relatively stable state, the freezing rate is almost zero, and the position of the freezing front also gradually tends to be stable. Comparing the moving distance of the freezing fronts of coal samples with different water contents, we can see that the larger the water content of the coal body, the faster the moving speed of the freezing fronts, and the farther the relative steady state moving position is from the freezing cold source.

The temperature gradient causes a slow decrease in the rate of freezing front motion, and the matrix potential within the coal is responsible for this change, too. However, the mechanism of the matrix potential within the coal causing this change is not clear, and the relevant equations and theoretical derivations are not clear enough.

Summarizing the above experimental data, it can be concluded that as the freezing process proceeds the temperature gradient gradually decreases, and the time required to lower the same temperature for the different water content coal samples becomes longer and longer, and when the freezing time is long enough, they will eventually all be lowered to an identical temperature, before which the different water content coal samples will reach a relatively stable temperature of their own and last for a longer period of time. This conclusion is consistent with the “three zones” theory, where the pore water in the coal starts to crystallize in the pre-freezing period and ice crystals are formed in the positive freezing zone. In the middle stage of freezing, most of the pore water phase becomes ice, and ice crystals are formed in large quantities at this time, and the area will gradually change to the frozen area. In these two processes, the temperature of coal sample continues to decrease. In the late freezing period, the volume of unfrozen area, positive frozen area and frozen area is stable and unchanged, and the temperature of coal sample reaches a relatively stable state. The movement law of freezing front can be analysed by deriving Darcy’s law of water permeation in coal and studying the influence of partitioning potential and matrix potential on it.

### 3.3. Analysis of the Mechanism of Water Migration during the Freezing of Coal Bodies

Due to the interaction between the coal matrix and its pore space and water, the water in the coal under freezing conditions does not freeze completely, but satisfies a dynamic equilibrium relationship with the temperature; i.e., the unfrozen water content decreases as the freezing temperature of the coal decreases, and there is always some free water present [30,31]. According to the “three zones” theory, the coal can be divided into three zones during freezing [32,33], namely, the unfrozen zone (>0 °C), the freezing zone (0 to −20 °C), and the frozen zone (<−20 °C). The freezing zone can be broadly defined as the freezing zone and the freezing edge, the interface between which is the ice lens, and the interface between the freezing edge and the unfrozen zone is the freezing front; i.e., the freezing edge corresponds to the freezing zone (0 to −20 °C). In his study of the relationship between freezing fronts and fracture production, Hall [34] found that fractures produced by freezing and swelling affect the direction of water migration, and that the production of fractures provides a transport channel for, and ultimately accelerates, water migration.

When the coal sample is exposed to a cold source falling below the freezing temperature, a very gentle zero isotherm forms within the coal and the pore water within the coal begins to crystallise, at which point ice crystals form in the positive freezing zone [35]. When the temperature within the coal gradually drops to the point at which the vast majority of the pore water phase becomes ice (considered to be completely frozen), at which point ice crystals form in large numbers, the area will gradually change to a frozen zone, and the freezing front will move outwards towards the cold source. As the freezing front moves, there exists a dynamic equilibrium between the water within the coal sample, the coal matrix, the structure of the ice–water interface, and the temperature gradient, and the ice lens body forms and gradually expands with the freezing front. As the temperature gradient increases, the freezing front continues to move, but the adsorption force of the lenticular body of ice is required to attract water molecules from the vicinity to its own surface to form a water film, from which new ice lenticular bodies are then first created, thereby enabling the freezing front to continue to move. However, the coal matrix in the vicinity of the freezing front pulls water molecules from the non-freezing zone to replenish the migrating water, which hinders the migration of water molecules to the ice crystals. The temperature gradient is the driving force behind water migration, and water molecules migrate towards the freezing front during freezing. A diagram of the migration of water molecules is illustrated in Figure 8.

From Figure 7, it can be seen that the temperature gradient causes a slow decrease in the speed of movement of the freezing front, while also asking what temperature gradient will have an effect on the speed of movement of the freezing front. According to the research of related scholars, it can be found that the temperature gradient affects the segregation potential during the low-temperature freezing of coal. The segregation potential affects the rate of water migration, but the exact mechanism of the effect, and the relationship between the segregation potential and the temperature gradient, is not clear.

Konrad et al. [36,37] studied the thickness of the freezing edge and proposed the concept of segregation potential, while Akagawa [38] studied the structural characteristics of the freezing edge, the growth rate of ice segregation, and its influencing factors. The percolation process of pore water is influenced not only by the hydraulic gradient and sorption, but also by the segregation potential (which refers to the temperature change that causes a change in the density of water and generates a certain pressure difference, which will lead to the flow of pore water in the density reduction direction), the solute potential, etc. [37,39]. The Darcy permeability equation has been studied by many scholars [40,41]; combining the experimental data from this paper, based on previous studies, the modified Darcy permeability equation for low-temperature coal can be expressed in pressure form as follows:(2)$$S\frac{\partial p}{\partial t}+\nabla \left[-\frac{k}{\eta }\nabla \left(p+\rho_{L}gH_{g}+\frac{SP_{0}}{k}T\right)\right]=Q_{H}$$
(3)$$\vec{u}=-\frac{k}{\eta }\nabla \left(p+\rho_{L}gH_{g}+\frac{SP_{0}}{k}T\right)$$
where S is the specific water storage coefficient, which is a constant, p is the permeation pressure, ∇ is the Hamiltonian operator, k is the permeability, η is the viscosity coefficient of water (0.001 kg/(m^−s^)), $\rho_{L}$ is the density of the flowing water, $H_{g}$ is the gravitational head height, $SP_{0}$ is the partial condensation potential coefficient (which is a positive constant at temperatures below freezing and zero at temperatures above freezing), T is the temperature, $Q_{H}$ is the source or sink of the seepage field, and $\vec{u}$ is the relative velocity vector of the fluid.

According to the experiment, it can be seen that the temperature gradient at the beginning of freezing is very large, and the coal body has a large density change due to the large temperature gradient, and the pressure caused by the density change is larger; i.e., the greater the partial segregation potential. As can be seen from Equation (3), it is clear that the larger the temperature gradient, the larger the value of temperature T and the larger the permeate pressure p. The larger the values of osmotic pressure and temperature, the larger the value of the relative velocity vector of water; i.e., the faster the migration rate of the water molecules, which is closely related to the movement of the freezing fronts. It can be concluded that the larger the temperature gradient, the greater the segregation potential and the faster the movement of the freezing front. Combining the equations and the experimental results of temperature propagation of coal under freezing conditions, it can be concluded that the variation law of the freezing front motion velocity is influenced by the segregation potential, and the change in the segregation potential is due to the change in the temperature gradient causing the change of the density of the coal. As the freezing process proceeds, the temperature gradient becomes smaller, the density remains constant, and the freezing front reaches a relatively stable state.

Unfrozen water in coal generally consists of bound water and free water; gravity water, as well as the capillary water in free water, will migrate towards the freezing zone during freezing, and only weakly bound water (sometimes called thin-film water) in the bound water will attach to the surface of ice crystals as a thin film and migrate. There are two main theories of water migration based on studies of unfrozen water migration, namely, the capillary water migration theory and the thin-film water migration theory [42]. While related theoretical studies have yielded theoretical models from the perspectives of water, ice pressure, and suction, no in-depth investigations have been carried out on the influencing factors of moisture migration and the initial coal water content, temperature gradient, etc. [43,44]. Moreover, most of the indoor model tests and numerical simulations that have been conducted were focused on the freezing of soils. For example, Nagare et al. [18]. conducted a two-way freezing test to study the effect of temperature on the redistribution of soil water potential and the water content. Shoop et al. [17]. The modified Darcy equation well explains the relationship between temperature gradient and deflection potential and verifies the motion law of the freezing front.

Whether in thawed or normal permafrost, water migration is caused by the different energies of water at different points in the soil; i.e., the soil water potential. It can be assumed that such a water potential also exists in coal rocks; strictly speaking, it is the total potential energy of the mineral components and water that constitute the coal rock. The various sub-potentials of the water potential of the coal rock play different roles in different situations. For non-water-filled rocks, the matrix and gravitational potentials play a dominant role [6]. The presence of unfrozen water during the freezing of a coal is mainly caused by the coal water potential; when the ice and unfrozen water reach equilibrium during the freezing of the coal, the temperature can be considered as the freezing point of unfrozen water. The gravitational potential of water in the coal is caused by the presence of the gravitational field, and is the work done by the water to overcome gravity by moving from the reference height *z*_0_ to the height *z*. In the actual calculation process, the reference height is set as the origin, at which the gravitational potential of water is zero (i.e., $z_{0}=0$). Then, if the coordinate axis is positive upwards, the gravitational potential g=z; if the coordinate axis is positive downwards, the gravitational potential g=−z. The matrix potential of the coal is caused by the suction and capillary forces on the water due to the porous structure inside the coal. The matrix potential of free water is set to zero. In the saturated zone, the matrix potential of moisture is comparable to that of free water, and the matrix potential $\varphi_{m\;}=0$; in the unsaturated zone, moisture must overcome matrix suction, and thus the matrix potential $\varphi_{m}<0$. Combined with experiment on the freezing front movement law of coal samples with different water content, the effect of water content on matrix potential was investigated by deriving the Platinum–Darcy law to reveal the mechanism of the influence of matrix potential on water migration.

In 1907, Edgar Buckingham modified Darcy’s law by extending it to represent the motion of water in a one-dimensional unsaturated porous medium in the vertical direction. The resulting equation is known as the Buckingham–Darcy law, which is expressed as follows:(4)$$q=-K(\theta )\frac{\partial \varphi }{\partial z}=-K(\theta )\frac{\partial \left(\varphi_{m}+z\right)}{\partial z}$$
where q represents the water flow velocity (cm/s), φ represents the total water potential in the vertical direction, which is equal to the sum of the matrix potential and the gravitational potential, $\varphi_{m}$ represents the matrix potential (m or cm), z represents the gravitational potential, which is positive in the upward direction (m or cm), and K(θ) represents the unsaturated hydraulic conductivity, which can be expressed as a function of the water content K(θ) or the matrix potential $K\left(\varphi^{m}\right)$ (cm/s).

The substrate potential $\varphi_{m}$ is negative, and Equation (4) indicates that the larger the substrate potential $\varphi_{m}$, the slower the water flow q and the stronger the impediment to water migration, and vice versa.

The absorption of the internal matrix potential during freezing becomes weaker with the increase of the water content, which is related to the structure of the coal rock. The relationship between the water content and matrix potential can be characterised by introducing a moisture characteristic curve model [45,46]. The following is an improvement and derivation of the moisture characteristic curve model combined with experimental data.
(5)$$\frac{\theta -\theta_{r}}{\theta_{s}-\theta_{r}}={\left[\frac{1}{1+h^{n}}\right]}^{m}$$
(6)m=1−1/n,   n>1
(7)$$S_{e}=\frac{\theta -\theta_{r}}{\theta_{s}-\theta_{r}},0<S_{e}\leq 1$$

The substitution of Equation (7) into Equation (5) yields the following:(8)$$h={\left[\frac{1}{{S_{e}}^{\frac{1}{m}}}-1\right]}^{\frac{1}{n}}$$
where θ denotes the initial water content, which is often expressed as a volume fraction (cm^3^/cm^3^), $\theta_{s}$ denotes the saturated water content (cm^3^/cm^3^), $\theta_{r}$ denotes the residual water content (cm^3^/cm^3^), h denotes the matrix suction (cm), i.e., the negative matrix potential, η is the shape parameter of the moisture characteristic curve, and $S_{e}$ denotes the effective water content; i.e., the degree of saturation.

Equations (7) and (8) show that the substrate suction h is influenced by the initial water content θ; the higher the initial water content θ, the lower the substrate suction h, and the lesser the impediment to water migration.

The effect of the initial water content θ on the water flux q, i.e., the rate of water migration, is derived by combining Equations (4) and (8). The greater the initial water content, the lower the matrix potential, the greater the water flux, and the faster the rate of moisture migration. Thus, the farther the freezing front moves when it reaches a relative steady state. As the residual water content of the coal sample gradually decreases during the freezing process, it leads to a gradual increase in the matrix potential within the coal. Therefore, the magnitude of the substrate potential is constantly changing during the freezing process, and the change trend is opposite to the change trend of the freezing front movement speed. Since the matrix potential acts as a barrier to the movement of the freezing front, Equations (4) and (8) verify the movement pattern of the freezing front.

## 4. Conclusions

In this research, the temperature field evolution law of coal under low-temperature freezing conditions was investigated via a real-time temperature measurement system. The temperature at different locations inside coal samples with different water contents was measured in real time, and the movement law of the freezing front and the moisture migration process were analysed. The following conclusions were obtained.
(1)Under the same low-temperature freezing conditions, the rate of temperature change at each measurement point inside the coal samples with different water contents was found to increase and then decrease until it reached zero. The closer the distance to the cold source, the faster the freezing front advanced, leaving less time for unfrozen water to migrate. Moreover, the faster the rate of temperature change, the faster the coal temperature field reached a relatively steady state.(2)The higher the water content of the coal during the freezing process, the faster the rate of temperature propagation. As the internal pore water phase of the coal changes to ice, the thermal conductivity of ice being much greater than that of water, the lower the temperature and the greater the thermal conductivity of ice; the volume expansion due to the change of the pore water phase into ice increases the contact area of the ice and the coal matrix, and the joint action of the two promotes the temperature propagation of the water-bearing coal.(3)The temperature field inside the coal after temperature stabilization undergoes a dynamic equilibrium process, and the temperature gradient is the driving force of moisture migration. The derivation of Darcy’s permeability equation reveals that the greater the temperature gradient, the greater the condensation potential, and the faster the internal water migration rate. During the low-temperature freezing process, the freezing front moves because the adsorption force of the ice lens body is greater than that of the coal matrix; this causes the water molecules adsorbed by the coal matrix in the unfrozen area to migrate to the freezing front and form a new ice lens body.(4)The −20 °C isothermal surface was defined as the freezing front of the coal, and the freezing front and freezing time change curves were plotted for each measurement point inside the coal. In combination with the theoretical analysis, it was concluded that the higher the initial water content, the smaller the matrix potential within the coal, the weaker the absorption of water molecules, the lesser the impediment to water migration, the faster the rate of water migration, and, when the relative steady state is reached, the longer the migration distance.

## Figures and Tables

**Figure 1 ijerph-18-13188-f001:**
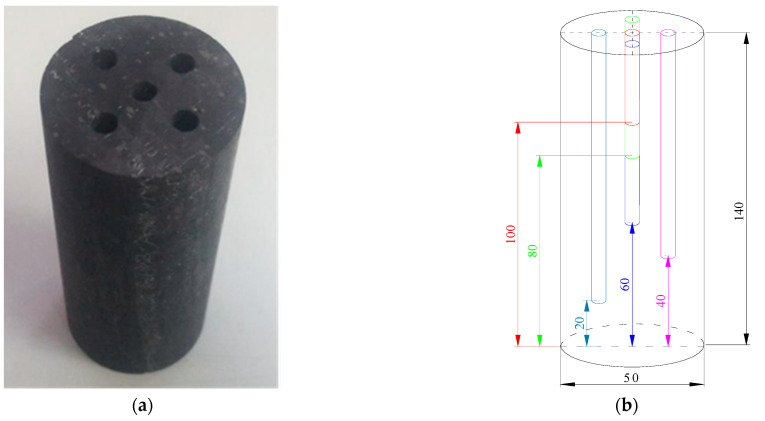
The test coal samples: (**a**) a photograph of a coal sample; (**b**) a diagram of coal sample drilling.

**Figure 2 ijerph-18-13188-f002:**
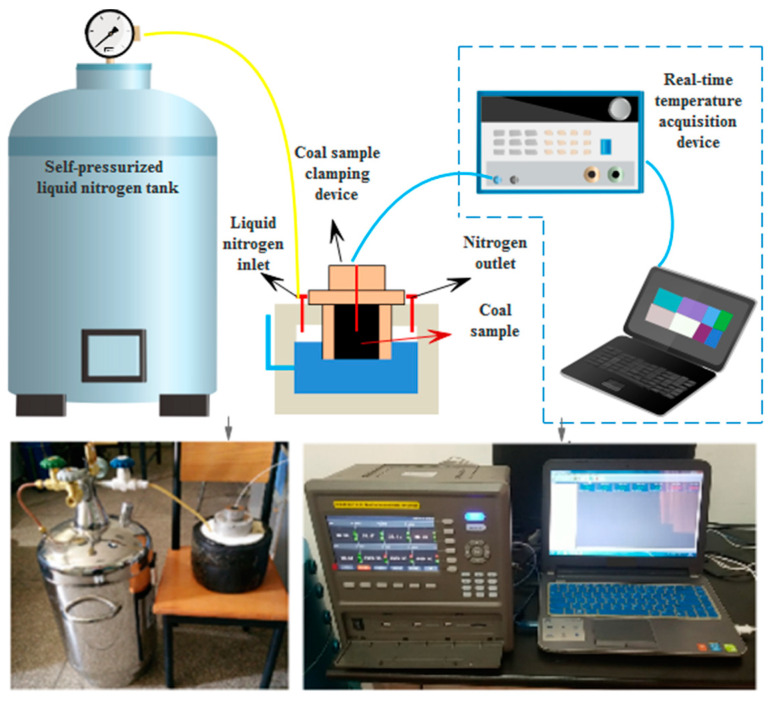
The schematic diagram of the real-time temperature measurement system for self-pressurised liquid nitrogen cold-soaking.

**Figure 3 ijerph-18-13188-f003:**
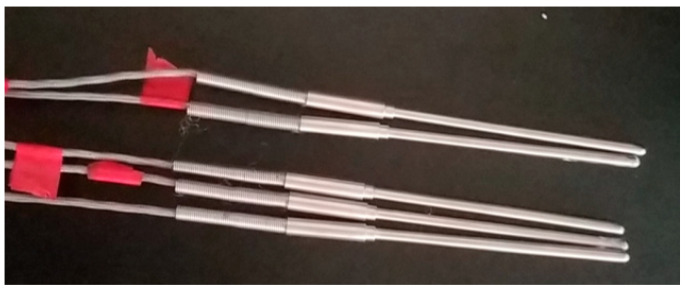
The three-wire PT100A platinum resistor.

**Figure 4 ijerph-18-13188-f004:**
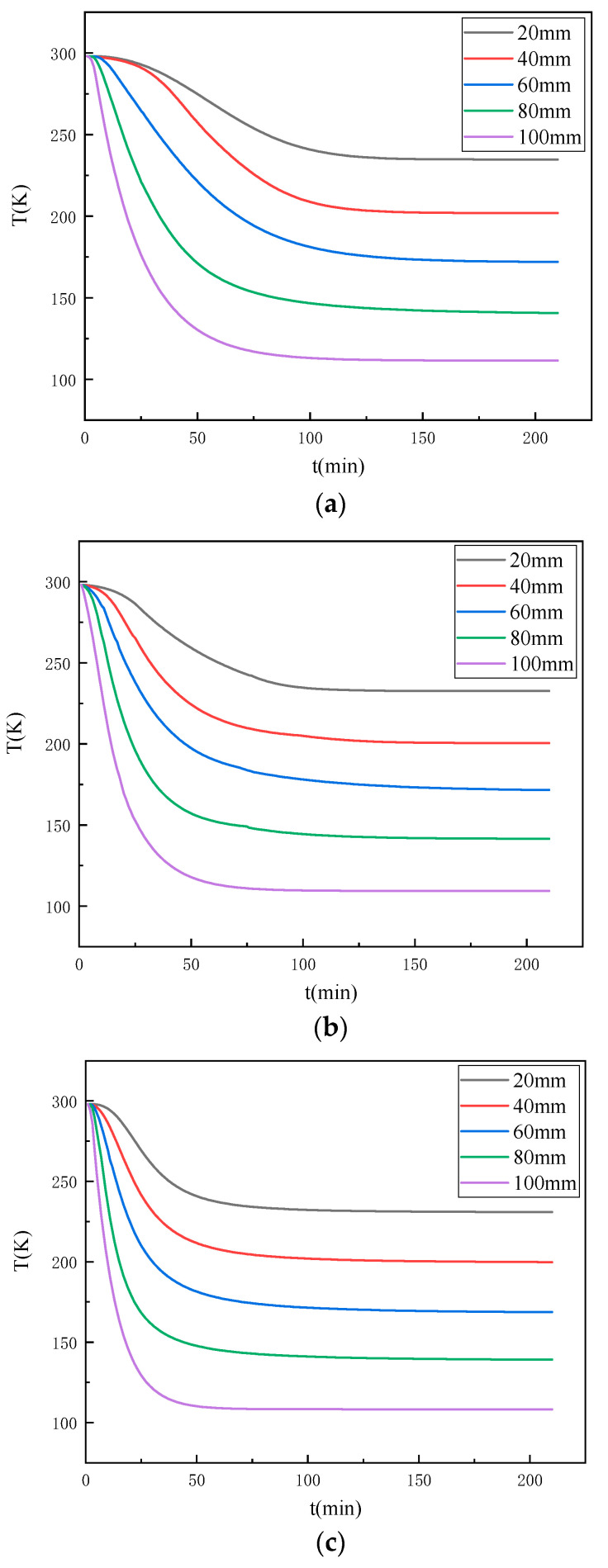
The temperature change at different points for coal samples with different water contents: (**a**) 0%; (**b**) 4.23%; (**c**) 8.76%.

**Figure 5 ijerph-18-13188-f005:**
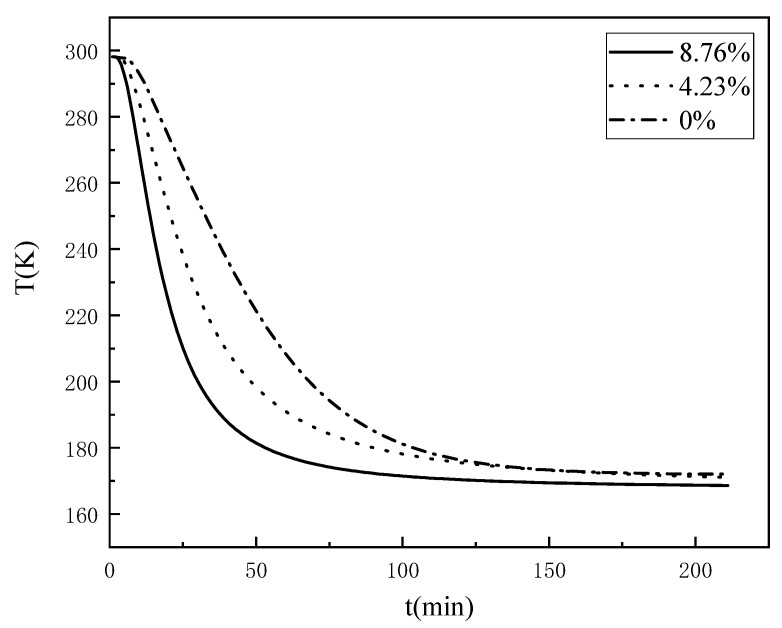
The law of temperature change under different water content conditions (60 mm).

**Figure 6 ijerph-18-13188-f006:**
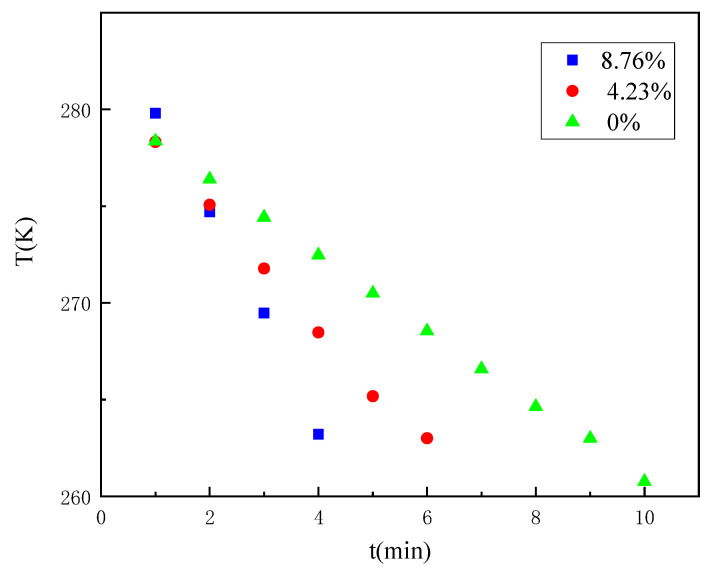
Temperature variation law of the coal samples with a different water content from 4 °C to −10 °C (60 mm).

**Figure 7 ijerph-18-13188-f007:**
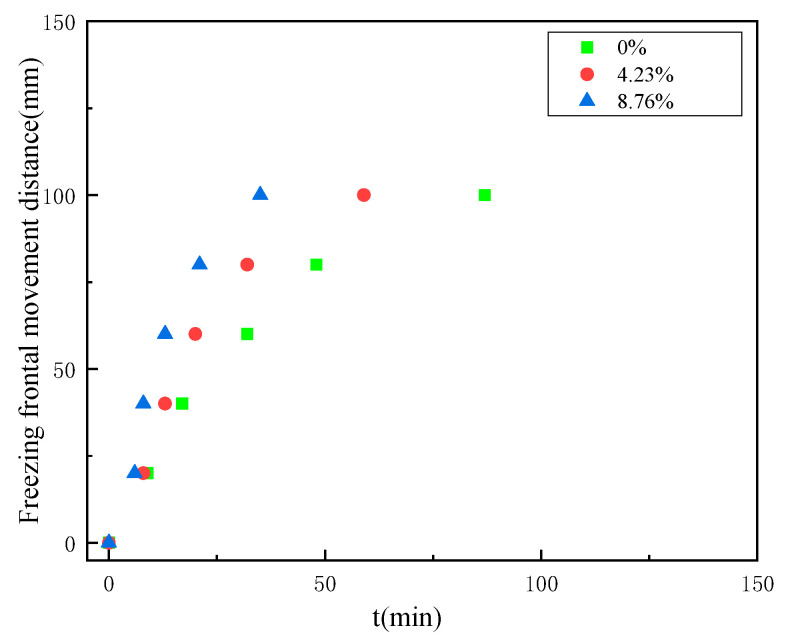
The movement of the freezing front under different water content conditions.

**Figure 8 ijerph-18-13188-f008:**
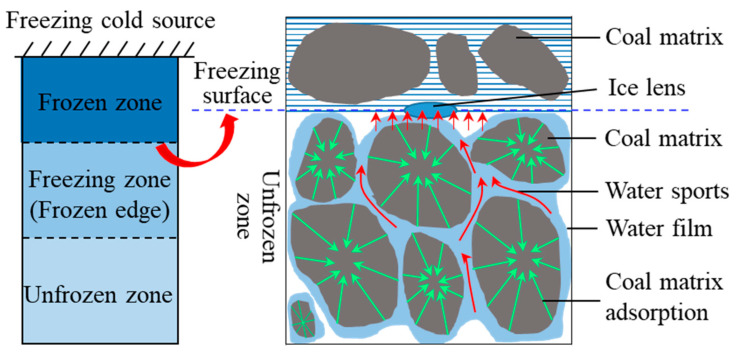
The flow of water molecules to the frozen area in coal.

**Table 1 ijerph-18-13188-t001:** Temperature variation law of the coal samples with a different water content from 4 °C to −10 °C (60 mm).

t (min)	Water Content	8.76%	4.23%	0%
1	T (K)	279.80032	278.32099	278.35563
2		274.69999	275.06943	276.39199
3		269.47052	271.77736	274.42745
4		263.21491	268.47143	272.46344
5		259.01736	265.17487	270.50139
6		253.94367	263.0076	268.54269
7		249.04281	258.68637	266.58876
8		244.34906	255.52502	264.64099
9		239.88443	252.43473	263.00075
10		235.66102	249.4243	260.76939

**Table 2 ijerph-18-13188-t002:** Variation law of temperature reduction of coal samples with a different water content from 4 °C to −10 °C (60 mm).

Water Content		8.76%	4.23%	0%
t (min)	Pre-phase	5.10033	6.54363	5.89219
	Mid-term	5.22947	6.59800	5.89060
	Late	6.25561	6.60249	5.88476

**Table 3 ijerph-18-13188-t003:** The variation pattern of the time required to lower the same temperature at the measurement points of coal samples with a different water content (60 mm).

T (K)	Water Content	8.76%	4.23%	0%
277–263	t (min)	4	6	9
263–249		3	5	9
249–235		4	6	9
235–221		5	8	9
221–207		6	10	12
207–193		10	15	17
193–179		21	40	31

## Data Availability

Not applicable.

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
