# Peer review of "The Temperature Field Evolution and Water Migration Law of Coal under Low-Temperature Freezing Conditions"

_ijerph, 2021, doi:10.3390/ijerph182413188_

Round 1

Reviewer 1 Report

Report on Manuscript no ijerph-1476701

General Overview

This manuscript is about the rate of temperature change versus distance from a cold source in  samples of anthracite which are dry or water saturated (and 50% saturated). The anisotropic rate of cooling is considered in terms of the water content of the samples. The methodology is quite simple in that it involves the placement of spaced-out sensors. The importance of the work is essentially about the structural integrity of coal under rapid cooling regimes. This sounds a deceptively simple thing to do but clearly it is not because there is always the general complication of the mode of measurement affecting the actual experiment. The cooling is from a silica ring which is liquid nitrogen cooled.   

There is no doubt that rapid changes in temperature are going to affect the overall structural stability of mineral samples so this is a meaningful exercise. After the results of the experiment are communicated there follows a discussion on the mechanism of the temperature transfer. This brings in many citations from related literature data. In this several equations which relate to the rate of cooling are presented. How do these relate to your data? I would have preferred a cogent discussion before the results are presented because I can’t see any evidence for the mechanism of heat transfer from the experimental data except for the generalisation that the higher the water content the faster the cooling. Looking at the fine detailed region of the curves between say 4oC and -10oC might tell you something about water phase change etc.

Overview

Firstly, there is some awkward phrasing in the manuscript which hides the meaning which can be tidied up with a good proof-reader.  The actual experiments carried out are clearly presented and the cooling curves make sense. I think that the data could also be presented in terms of a simple graph time to reach a certain temperature for all three samples at fixed distances. I think duplicate experiments would be good. The part of the manuscript that I got lost on was how the work here related to the equations. As far as the equations are concerned, I would define each term relating these to their physical meanings otherwise these are just random equations from the literature on related subjects. Also, where there are mathematical operators in the equations (usually arising from complex matrix algebra) define their role to the reader. As far as the mechanism of propagation of what I’ll call the cold front, how do you know the role of water in heat dissipation? When the curves level out, I guess this means the rate of cooling is equal to the rate of warming hence the system is in equilibrium- is this correct?  You state there is an equilibrium but don’t expand on this.

 I have some very simple questions which are required to be addressed by the authors before this work can be accepted.

These are as follows:

  1. Sensors: do these record the temperature at the tip (as stated) only which is clearly required for any meaningful results. Although this is stated in the experiment but has this been tested by having two different cooling points along the wire. The contact point and area of contact is also vital Also, if the sensors are in the sample and clearly metallic do these affect the temperature changes in the bulk sample as they are conduction channels. Why do you not use temp. in K ?
  2. Are there duplicate runs available or another sample. Some sort of averaging of results may be required. How reproducible are the results?
  3. When the temperature change passes through the water freezing range would you not expect some curve deviation based on latent energy transfers. This may be there but is buried in the detail.
  4. In the equations presented what are the meanings of the numerical values (fit coefficients) calculated for the exponential temperature drop curves and how do these relate to the equations presented? Where does equation 5 come from in the first place? The curves in figure 6 look like simple ln curves (first order rate curves). Did you try to model curve based on this? It would seem the obvious choice (the same way a least-squares fit is done when you have y = c1+c2, wy = c1+c2w, but in place of the w values you put the values of ln x.)
  5. Is the external humidity controlled as cooling will suck in moisture from the atmosphere?
  6. When 50% of the water is removed from a sample how do you know the sample is homogeneous as one might think water will come from initially from the extremities and equilibration will take a very long time.
  7. Are the drilled holes completely independent of one another under the cooling conditions? You could drill a sample with equal depths and look at the simultaneous results from each of the sensors.

To be considered for publication I would present all the relevant theory from other workers first in the general introduction and then relate your results to the literature work. At present its difficult to see what is your work and what is the literature work. I realise you make use of lit. data such as three zone system and employ that approach, but you should rationalise why you are doing this. Overall, the paper can be simplified I think without the need for reference to many of the equations cited from elsewhere.

Reviewer 2 Report

The main question addressed by the research is he water migration inside coal under freezing conditions. Derivation of Darcy's law for water permeation in coal under those conditions.  

The manuscript is well written and the content is scientifically sound.

The topic is relevant for materials science and construction materials and of application in building/infrastructure construction in cold regions. A better explanation why the authors selected coal as test material would be an improvement. Also the discussion how the results may apply to other materials would help. The references are appropriate and tables and figures are well presented.

However, the content although relevant to engineering has no clear relationship (and no discussion and no conclusions) with  Public Health.

Authors could increase the impact and readership of this manuscript through publishing in another journal more related to materials and engineering.

Round 2

Reviewer 1 Report

Please see short report.
